



# Technical note: Facilitating the use of low-cost methane (CH₄) sensors in flux chambers – calibration, data processing, and an open source make-it-yourself logger

David Bastviken[1], Jonatan Nygren[1], Jonathan Schenk[1], Roser Parellada Massana[1], Nguyen Thanh Duc[1]

[1]Department of Thematic Studies – Environmental Change, Linköping University, 58183 Linköping, Sweden

*Correspondence to*: David Bastviken (david.bastviken@liu.se)

**Abstract.** A major bottleneck regarding the efforts to better quantify greenhouse gas fluxes, map sources and sinks, and

understand flux regulation, is the shortage of low-cost and accurate-enough measurement methods. The studies of methane
($CH_4$) – a long-lived greenhouse gas increasing rapidly but irregularly in the atmosphere for unclear reasons, and with poorly
understood source-sink attribution – suffer from such method limitations. This study present new calibration and data
processing approaches for use of a low-cost $CH_4$ sensor in flux chambers. Results show that the change in relative $CH_4$ levels
can be determined at rather high accuracy in the 2 – 700 ppm range, with modest efforts of collecting reference samples *in*

*situ*, and without continuous access to expensive reference instruments. These results open for more affordable and time-
effective measurements of $CH_4$ in flux chambers. To facilitate such measurements, we also provide a description for building
and using an Arduino logger for $CH_4$, carbon dioxide ($CO_2$), humidity, and temperature.

## 1 Introduction

Methane ($CH_4$) is the second most important of the long-lived greenhouse gases (GHGs). Its global 100-year warming

potential per mass ($GWP_{100}$) is 28-34 times greater than the $GWP_{100}$ for carbon dioxide ($CO_2$), and its relative increase in the
atmosphere since 1750 have been much greater than for other GHGs (e.g. Myhre et al., 2013). The atmospheric $CH_4$
originate from multiple sources including incomplete combustion, natural or biogas gas handling, or microbial $CH_4$
production in agriculture, ruminant digestive tracts, and other anaerobic environments such as wetlands and lakes – the
microbial $CH_4$ accounting for approximately two thirds of the total emissions (e.g. Saunois et al., 2016). The high diversity

of sources, many yielding fluxes that have high spatio-temporal variability, makes it difficult to quantify fluxes and
understand flux regulation without a large number of local measurements. At the same time, common methods to measure
fluxes rely on expensive equipment or labour-demanding procedures. Consequently, the $CH_4$ flux from various sources are
poorly constrained. This is exemplified by the discovery of inland waters and flooded forests as two large global $CH_4$
sources during the last decade (Bastviken et al., 2011;Pangala et al., 2017).  Greater availability of measurement approaches

that are inexpensive enough to allow many measurements and assessment of both spatial and temporal variability
simultaneously, would greatly improve our ability to assess landscape $CH_4$ fluxes and flux regulation.

There is extensive work to develop sensitive, small, and affordable $CH_4$ sensors, but so far the commercially available low-cost $CH_4$ sensors were typically developed for explosion warning systems and thereby for high concentrations (percent

levels). $CH_4$ detection at such levels is of high interest for environmental research, including the measurements of $CH_4$ ebullition, and for such applications cost-efficient sensor applications have been presented (e.g. Maher et al., 2019). For measurements of other types of $CH_4$ fluxes, sensors with robust and reliable detection at lower levels (ppm) are needed. Previous attempts to use and calibrate such sensors at low levels have been promising (e.g. Eugster and Kling, 2012; see Table S2), but also reported remaining challenges, and the use of these sensors in environmental research or monitoring has

not yet become widespread. The direct monitoring of atmospheric $CH_4$ levels to resolve fluxes, demanding fast and accurate detection of changes in the order of 10ppb, still represents a challenge for low-cost sensors. However, relevant level ranges for flux chamber studies (2 – ~1000 ppm depending on environment, chamber type, and deployment times) appear within reach.

One  commercially available low-cost sensor type, showing promising performance in previous studies, are represented by the TGS 2600 tin dioxide ($SnO_2$) semiconductor sensor family made by Figaro. This type of sensors has been evaluated multiple times at $CH_4$ levels near ambient background air (from 1.8 to 9 ppm; different ranges in different studies; Eugster and Kling, 2012; Casey et al., 2019; Collier-Oxandale et al., 2018; van den Bossche et al., 2017). Given their low cost, they performed surprisingly well under non-sulphidic conditions ($H_2S$ may interfere with the sensors), although it was

challenging to generate calibration models with $R^2 > 0.8$, and the reported interferences from e.g. relative humidity (*RH*) and temperature (*T*) were large (van den Bossche et al., 2017; Table S2). We here evaluate one member of this sensor family for a larger $CH_4$ range (2-719 ppm), selected to be appropriate for use in automated and manual flux chambers. We propose further development of the equations suggested by the manufacturer for data processing, and provide guidance on how to address the sensor response to humidity (*H*), *RH* and *T* in flux chamber applications. We also describe a simple

$CH_4$/$CO_2$/*RH*/*T* logger based on the evaluated sensors, an Arduino microcontroller, and a corresponding logger shield.

## 2 Methods

### 2.1 The $CH_4$ sensor

The sensor used in this study is the Figaro NGM 2611-E13, which is a factory pre-calibrated module based on the Figaro TGS 2611-E00. The factory calibration is made at 5000 ppm, 20 °C and 65% *RH*. These levels are is not relevant for

applications near atmospheric concentration or variable *T* and *RH*, but the NGM 2611-E13 is compact and ready-to-use, facilitating its integration with data loggers and equipment for flux measurements (eg. automated flux chambers; Duc et al., 2013; Thanh Duc et al., 2019). The detection range given by the manufacturer is 500-10000 ppm, but the sensor has been used successfully for measuring indoor ambient concentrations of methane (van den Bossche et al., 2017). The potential of

another similar sensor, the Figaro TGS 2600, for atmospheric concentration monitoring have been investigated (Eugster and

Kling, 2012; Collier-Oxandale et al., 2018). The main difference of the TGS 2611-E00, compared to the TGS 2600, is the presence of a filter that reduce the interference of other combustible gases with the sensor, making it more selective towards $CH_4$ (Figaro_TGS_2611-E00, REV: 10/13). The TGS 2611-E00 is also more than 10 times cheaper than the sensor used in Duc et al. (2013) and its detection range is wider, allowing for reliable measurements of concentration above 1000 ppm, which makes the sensor potentially useful in both low- and high-emitting environments.

**2.2 Calibration setup**

The sensor evaluation set-up was designed to resemble real measurement conditions in floating flux chambers in aquatic environments. The sensors were placed in the headspace of a plastic bucket positioned upside down on a water surface in a tank. We used a 7L plastic bucket in which we located 20 TGS 2611-E13 sensors connected to electronic circuitry and sensor signal logging system described in detail separately (Thanh Duc et al., 2019). The chamber headspace was

continuously pumped from the chamber, through the measurement cell of an Ultraportable Greenhouse Gas Analyzer (UGGA; Los Gatos Research), and then back to the chamber. The UGGA served as a reference instrument for $CH_4$. The air $T$ and $RH$ inside the chamber were measured with ten K33-ELG sensors which have an accuracy of ± 0.4°C and ± 3% $RH$ (Senseair; also used for $CO_2$-measurements and therefore suitable for simultaneous use in flux chamber measurements; Bastviken et al., 2015; data from two K33-ELG sensors and four $CH_4$ sensors were logged together). The entire installation

was placed in a climate room to allow for varying $T$, and thereby also absolute humidity ($H$) in the chamber headspace. $T$ and $H$ co-vary under field conditions in measurements near moist surfaces, so although $T$ and $H$ were not controlled independently, their variability under this calibration setup was reflecting flux chamber headspace conditions under in situ field conditions.

The $CH_4$ concentration in the chamber was changed by direct injections of methane into the chamber by syringe via a tube. The $CH_4$ concentrations during the calibration experiments ranged from 2 ppm to 719 ppm. We performed multiple separate calibration experiments at different $T$ and $RH$ levels ranging from 10 to 42 °C and 18 – 70 %. Values were recorded once per minute. $T$ and $RH$ values form the $CO_2$ sensors were averaged among all sensors before used in the evaluation.

The response time to changing chamber headspace $CH_4$ levels differed between the sensors situated in the chamber (responding rapidly), and the UGGA (delayed response time due to the residence time of the measurement cell and tubing). Therefor data were filtered to remove periods of rapid changes when the different response times caused data offsets. Some sensor data were lost during parts of the experiments due to power, connection failure, or data communication issues. Altogether on an average, after data filtration, 619 – 930 data points from each sensor and the UGGA, respectively, were

used for the evaluation (in total 20 $CH_4$ sensors evaluated).



## 2.2 Data processing and interpretation

The TGS 2611 $SnO_2$ sensing area exhibit decreasing resistance with increasing methane concentration (Figaro_Tech_Info_TGS2611, 2012). The sensing area is connected in series with a reference resistor (resistance referred to as $R_L$). The total circuit voltage ($V_C$) is 5V across both the sensing area and the reference resistor. The voltage across the reference resistor ($V_L$) therefore varies in response to how the sensing area resistance ($R_S$) varies. $V_L$ is measured and reported as output voltage. The sensor response $R_S$ is calculated from the following equation (Figaro_Tech_Info_TGS2611, 2012; Figaro_TGS_2611-E00, REV: 10/13):

$$R_s = \left(\frac{V_C}{V_L} - 1\right) \times R_L \qquad (1)$$

The active sensor surface characteristics and $R_L$ can differ among individual sensors, which makes individual sensor calibration necessary. Interference by water vapour and $T$ has been previously shown (Pavelko, 2012; van den Bossche et al., 2017). $R_L$ is therefore ideally determined in dry air containing no volatile organic compounds or other reduced gases at a standard $T$. However, it can be challenging to determine $R_L$, and Eugster and Kling (2012) proposed to use the lowest measured sensor output voltage ($V_0$), representing minimum background atmospheric levels, to determine an empirical reference resistance $R_0$, and to calculate the ratio of $R_S/R_0$, reflecting the relative sensor response as follows:

$$\frac{R_S}{R_0} = \frac{\left(\frac{V_C}{V_L} - 1\right)}{\left(\frac{V_C}{V_0} - 1\right)} \qquad (2)$$

This approach allows sensor use without accurate specific determination of $R_L$. Previous attempts to calibrate these type of sensors for environmentally relevant applications have focused on $CH_4$ levels of 2-9 ppm, and typically considered the influence of $T$ and $RH$ or $H$ (Casey et al., 2019; van den Bossche et al., 2017; Collier-Oxandale et al., 2018; Eugster and Kling, 2012). In these cases, an approximately linear response of the relative sensor response could be assumed due to the narrow $CH_4$ range. However, the sensor response is non-linear in the range relevant for flux chamber measurements and in this wider range, other approaches are needed. We here present a two-step sensor calibration based on the complete calibration experiment data. In addition, we tried simplified calibration approaches for situations when full calibration experiments are not feasible and when access to reference instruments is limited. These approaches are described below.

### 2.2.1 Two-step calibration from complete experimental data (Approach I)

The first step (Step1) regards determination of the reference sensor resistance, $R_0$. We assumed that $R_0$ represented $R_L$ + $R_{Sbkg}$, where $R_{Sbkg}$ is $R_S$ at the background atmospheric $CH_4$ level. We first tried the previously suggested approach to determine $R_0$ from the minimum $V_L$, i.e. setting $V_0$ to $V_L$ at the lowest humidity and $CH_4$ concentrations during all





measurements, thereby assuming that $R_0$ could be seen as constant. However, $R_{Sbkg}$ may be influenced by $H$ and $T$ and could vary even if the $CH_4$ levels at background atmospheric conditions are constant. Thus, we also tested ways to correct $R_0$ to $RH$ or $H$ and $T$. Therefore, after selecting the experiment data at background $CH_4$ levels but variable humidity and temperature,

we tested linear, power, or Michaelis-Menten models, to generate $V_0$ values valid for different humidity and temperatures. This allowed estimation of $R_0$ values at the humidity and temperature associated with each $R_S$ value, making the $R_S/R_0$ ratio less biased. The background level $CH_4$ data was selected in two different ways – either as all known $CH_4$ values below 2.5 ppm (n = 38-72), or as the minimum $V_L$ value for each experiment and sensor (n = 6-7).

The second step (Step 2) regards calculation of $CH_4$ levels from $R_S/R_0$. Several models were tested, where the $CH_4$ levels were estimated as a function of $R_S/R_0$, humidity, temperature, and a constant to consider offsets that may differ among sensors. We tried several linear and power functions. In line with viewing the sensor surface as an active site where $CH_4$ and $H_2O$ compete for space, the humidity effect was in some models represented as an interaction with the sensor response. In all above cases, models were generated by curve-fitting in Python using the scipy.optimize curve_fit function. Predicted

$CH_4$ levels were evaluated by comparison with the observed levels (independently measured by the UGGA). The specific model equations are provided in Table 1 and 2. We tested models using $RH$ or $H$ (which was calculated from $RH$ and $T$; (Vaisala_Technical_Report, 2013). Each evaluation included a combination of both steps above, and generated one set of fitted parameters per sensor used, including the parameters for Step 1 and 2.

### 2.2.2 Simplified calibration approaches without dedicated calibration experiment data (Approach II and III)

The model combinations from Step 1 and 2 above that generated the best fit with the minimum number of parameters was selected for tests of two simplified calibration approaches. In *Approach II* we tested if model parameters in Step 2 can be predicted from parameters derived in Step 1, hypothesising that the derived model parameters in both Step 1 and Step 2 reflect the sensor capacity to respond to $CH_4$ and humidity levels as well as the individual sensor offset. If correct, the parameters in Step 1 should be correlated with parameters in Step 2. If this correlation is strong enough, it may be possible to

predict parameters in Step 2 from parameters in Step 1, which can be derived from measurements at background air concentrations under the natural variation in humidity (e.g. the diel variability), as a part of the regular measurements, preferentially using data when the atmospheric boundary layer is well mixed (e.g. windy conditions). Under such conditions atmospheric background $CH_4$ concentrations can be relatively accurately assumed. Hence this Approach II would not require access to sensor calibration chambers nor expensive reference gas analysers, which in turn would make sensor measurements

available much more broadly. To test this approach, we searched for the best possible regression equations to predict Step 2 parameters from Step 1 parameters, then used these equations to estimate $CH_4$ levels, and compared estimated levels versus observed.





In *Approach III* we evaluated if reasonable accurate Step 1 and 2 equations can be derived from the combination of (*i*)
minimum background atmospheric level $V_L$ at different humidity, and (*ii*) a limited number of randomly collected
independent manual flux chamber samples. If so, a few manual samples during the regular measurements could replace
tedious dedicated calibration experiments. To test this approach the calibration data for each sensor was subsampled
randomly and this random subset data were combined with the minimum $V_L$ data to derive calibration parameters as done in
Approach I. Using these parameters, the $CH_4$ levels for the entire calibration data was estimated and compared with observed
values. Monte Carlo simulations were run to test effects of the number of random reference samples (1 – 50) and the
methane concentration ranges (3 – 500 ppm, or 3-50 ppm, respectively) in the subset data.

### 2.2.3 A low-cost Arduino-based $CH_4$/$CO_2$/*RH*/*T* logger

To facilitate use of the sensors and our results, we also gathered instructions for how to build a logger for $CH_4$, $CO_2$, *RH* and
*T* measurements, using the $CH_4$ sensor tested here, and the Senseair K33 ELG $CO_2$/*RH*/*T* sensor described elsewhere
(Bastviken et al., 2015), a supplementary DHT22 sensor for *RH* and *T*, an Arduino controller unit, and an Adafruit Arduino
compatible logger shield with a real time clock (Figure 2). This development was based on sensor specifications and the
open source knowledge generously shared on internet by the Arduino user community. The full description of this logger
unit is found in the Supplement.

### 3 Results and Discussion

The results of different Step 1 and Step 2 calibration equations are shown in Tables 1 and 2. The models including *H* were
equal or superior to models using *RH*. This is reasonable because it is the absolute water molecule abundance that influence
the sensor response. Hence, models using *H* were prioritized. In *Approach I*, several Step 1 models, including a constant
minimum $V_L$, and power, linear and Michaelis-Menten-based equations gave similar $R^2$ (0.85 to 0.9) and root mean square
error (RMSE) when comparing predicted versus observed results (Table 1). The effect of *T* appeared negligible compared to
*H*, which may be related to the built-in heating of the active sensor surface. It is possible that the Michaelis-Menten equation
is superior over the full theoretically possible *H* range. However, under our experiment conditions, covering normal field *H*
levels, the combination of best fit and minimum number of parameters in Step 1, was found for a simple linear equation with
*H* (Model V4 in Table 1), which was used for later tests of *Approach II* and *III*.

The tests of different equations in Approach I, Step 2, showed that power relationships with *H* and *T* represented as
interactions with the sensor response, performed best (Table 2 Model ≥4). With the exception of Models 10a-c, all these
models had in the regression of observed versus predicted, a slope and intercept that was statistically indistinguishable from
1 and 0, respectively ($p < 0.05$) and an $R^2$ of 0.98 – 1.00 (Table 2, Figure S1). Again, *T* had a marginal effect and *H* was
clearly most important. Hence, while Model 7 including *T* in Table 2 had the lowest RMSE (9.8), Model 8 represented a

good compromise between minimum number of parameters and low RMSE (10.4) and was used in Approach II and III. The
        non-linear response of the sensor yielded a stronger and more coherent response at low $CH_4$ levels, and a large part of the
        uncertainty were generated at the higher $CH_4$ levels in the studied range (Figure S1). Near the atmospheric background at 2
        ppm, the confidence interval for individual sensor response was in the order of ± 1.1 ppm (Model 7 having lowes RMSE).
        Hence, the presented calibration equations have a limited accuracy in terms of absolute $CH_4$ levels, and is not optimized for
high-precision measurements at atmospheric background levels. However, results indicate that the relative change of $CH_4$
        levels over time, which is the core of flux chamber measurements, can be assessed efficiently with the sensors if calibrated
        properly.

        *Approach II*, deriving all calibration equations from a small set of minimum $V_L$ values using Models V4 or V5 (Table 1) and
10a-c (Table 2), generated substantially greater RMSE. Most of this RMSE change was due to less accurate prediction of the
        intercept. The $R^2$ and slope standard error range remained similar to the other models (Table 2), but the actual slope values
        could deviate substantially from 1 and varied considerably among sensors (in contrast to the models for all other approaches
        always having slopes close to 1 and similar among sensors; Figure S1). Thereby, Approach II could lead to a large bias in
        absolute levels, and this crude generation of calibration equations may be adequate primarily for assessing relative change
over time measured by the same sensor, and cross comparisons among sensors should be avoided when using this approach.
        Examples of equations for the parameter estimation in Approach II is provided in Table S1. Applying Approach II on a
        smaller concentration range yielded a considerably lower RMSE (Table 2, Model 10c).

        *Approach III* (Model 11a and 11b in Table 2) showed that as few as 20 reference samples, collected at random occasions
during actual measurements, could substantially reduce the RMSE of the calibration models, reaching close to the lowest
        levels based on the 619 – 930 measurements and the full range up to 719 ppm in Approach I (Table 2 Model 11a; Figure S2).
        The concentrations of the reference samples did not appear important for the RMSE within a given specific data range.
        However, simulations using data for $CH_4$ levels below 50 ppm only, generated much lower RMSE than using all data (Table
        2, Model 11b). This support the conclusion that the sensors are more sensitive and give a stronger relative response in the
low part of the studied concentration range.

        An overview of approaches to derive calibration models for this type is shown in Table S2. The challenges found regarding
        monitoring of background atmospheric levels was confirmed by our study, while use for relative changes of greater
        magnitudes in flux chambers appear promising based on this study, also with a simplified calibration (Approach III).

**4 Conclusions**

        The main conclusions can be summarized by the following:

- The tested CH$_4$ sensors are suitable for use in flux chamber applications if there are simultaneous measurements of relative humidity and temperature (or humidity).

- Sensor-specific calibration is required.

• Occasional independent reference samples during regular measurements, is an alternative to designated calibration experiments. Background atmospheric levels in combination in the order of 20 *in situ* reference samples at other CH$_4$ levels, can yield rather accurate calibration models.

- For highest accuracy regarding absolute CH$_4$ concentrations, careful designated calibration experiments covering relevant environmental conditions are needed.

• These results, together with the increased accessibility of low-cost sensors and data logger systems (one example described in the Supplement), open supplementary paths toward improved capacity for greenhouse gas measurements in both nature and society.

**5 Code and data availability**

Python code for data evaluation and the calibration experiment data is available from the main author upon request. Please
note that both the code and the data are specific for the experimental setup. The Python code needs modifications for use with other data, and the CH$_4$ sensor data cannot represent results from other sensors as sensor specific calibration is needed.
The Arduino code for the CH$_4$/CO$_2$/*RH*/*T* logger described in the supplement is available at
http://urn.kb.se/resolve?urn=urn:nbn:se:liu:diva-162780.

**6 Author contribution**

DB and NTD designed and supervised the study. NTD designed and built the experimental setup, the sensor units and the data logging system. JN and NTD performed the calibration experiment. The analysis of the sensor data was led by DB with contributions from NTD, JS, and JN. DB and NTD developed the logger units presented in the supplementary material. RP helped to build and test logger units and drafted a user manual. DB wrote the first complete draft of the manuscript and led
the manuscript development, with contributions from NTD, JS, RP, and JN.

**7 Competing interests**

There are no competing interests.



## 8 Acknowledgements

This work was funded by the European Research Council (ERC) under the European Union's Horizon 2020 research and
innovation programme (grant agreement No 725546), and the Swedish Research Councils VR and FORMAS (grant no.
2016-04829 and 2018-01794, respectively). We thank Sivakiruthika Balathandayuthabani, Magnus Gålfalk,
Balathandayuthabani Panneer Selvam, Gustav Pajala, David Rudberg, Henrique Oliveira Sawakuchi, Anna Sieczko, Jimmy
Sjögren, and Ingrid Sundgren, for stimulating discussions regarding sensor use.

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



**Table 1:** Model results for Step 1 of sensor calibration - i.e. the correction of reference output voltage ($V_0$ in the unit mV) in background
air to humidity and temperature. $V_{0min}$, $H$, and $T$, represent the minimum $V_0$ for each sensor (mV), absolute humidity (ppm), and
temperature (°C) during measurements in open air. The model parameters $g$, $h$, $S$, $m$ and $n$ are constants for each sensor derived by curve
fitting. The model $R^2$ is the adjusted coefficient of determination (mean, minimum and maximum for the 20 sensors tested), and RMSE is
then root mean square error. Equivalent models using relative humidity ($RH$; %) instead of $H$, returned lower $R^2$ and higher RMSE and are
not shown. These Step 1 models were combined with the Step 2 models as noted in Table 2. See text for details.

| **Model for $V_0$** | | **Data** | **n** | **Observed vs. Predicted** | | | |
| **No.** | **Equation** | | | **$R^2$** | | | **RMSE** |
| | | | | mean | min | max | mean |
| V1 | $V_{0min}$ (constant) | Minimum $V_0$ | 1 | - | - | - | - |
| V2 | $V_0 = gH^h + mT^n + S$ | All < 2.5 ppm CH$_4$ | 38 - 72 | 0.85 | 0.66 | 0.94 | 8.9 |
| V3 | $V_0 = gH + mT + S$ | All < 2.5 ppm CH$_4$ | 38 - 72 | 0.88 | 0.68 | 0.95 | 8.2 |
| V4a | $V_0 = gH + S$ | All < 2.5 ppm CH$_4$ | 38 - 72 | 0.88 | 0.68 | 0.95 | 8.2 |
| V4b | $V_0 = gH + S$ | Min $V_0$ for each exp. | 6 - 7 | 0.90 | 0.72 | 0.96 | 8.3 |
| V5a | $V_0 = gH / (S + H)$ | All < 2.5 ppm CH$_4$ | 38 - 72 | 0.88 | 0.70 | 0.96 | 8.0 |
| V5b | $V_0 = gH / (S + H)$ | Min $V_0$ for each exp. | 6 - 7 | 0.89 | 0.71 | 0.96 | 8.3 |






**Table 2:** Model results for Step 2 of the data evaluation, i.e. the determination of methane ($CH_4$) levels (ppm) from the sensor response expressed as $R$ (corresponding to $R_S/R_0$) using different calibration models. $(RH)$, $H$, and $T$ as defined in Table 1. The model parameters $a$, $b$, $c$, $d$, $e$, $f$ and $K$ are constants for each sensor derived by curve fitting. The models were evaluated via a linear regression of *Observed* versus *Predicted* $CH_4$ levels, where $k$ and $M$ are the slope and the intercept, respectively. SE denote standard error, $R^2$ the adjusted coefficient of determination (mean and minimum to maximum for the 20 sensors tested), and RMSE is the root mean square error. The

table show the most successful subset of all models tested. N = 619 – 930 per sensor in total and 203-313 for the data subset with CH4 levels < 50 ppm. See text for details.

| Model | | $V_0$ | $CH_4$ | \multicolumn{5}{c}{$Observed = k * Predicted + M$} | | | | |
|---|---|---|---|---|---|---|---|---|
| No. | Equation | mod | max | $k$ | $M$ | | $R^2$ | RMSE |
| | | | ppm | SE (min-max) | min to max* | SE (min-max) | mean (min-max) | mean (min-max) |
| 1 | $CH_4 = aR + b(RH) + cT + K$ | V1 | 719 | 0.024-0.036 | $-3.2·10^{-7}$ to $3.5·10^{-7}$ | 5.8-8.2 | 0.58 (0.54-0.68) | 117 (104-127) |
| 2 | $CH_4 = aR^b + c(RH)^d + eT^f + K$ | V1 | 719 | 0.006-0.010 | $-8.8·10^{-7}$ to $4.2·10^{-7}$ | 1.6-2.6 | 0.96 (0.94-0.97) | 35.9 (32-45) |
| 3 | $CH_4 = aR^b + c(RH)(aR^b) + dT(aR^b) + K$ | V1 | 719 | 0.003-0.006 | $-6.8·10^{-7}$ to $9.3·10^{-7}$ | 0.72-1.44 | 0.99 (0.98-0.99) | 18.5 (15-25) |
| 4 | $CH_4 = aR^b + cH(aR^b) + dT(aR^b) + K$ | V1 | 719 | 0.002-0.003 | $-4.3·10^{-7}$ to $3.2·10^{-7}$ | 0.43-0.90 | 1.00 (0.99-1.00) | 11.4 (9-16) |
| 5 | As No. 4 | V2 | 719 | 0.001-0.003 | $-3.3·10^{-7}$ to $4.1·10^{-7}$ | 0.38-0.87 | 1.00 (0.99-1.00) | 10.6 (8-16) |
| 6 | As No. 4 | V3 | 719 | 0.001-0.003 | $-4.1·10^{-7}$ to $3.6·10^{-7}$ | 0.37-0.82 | 1.00 (0.99-1.00) | 9.8 (8-15) |
| 7 | As No. 4 | V4a | 719 | 0.001-0.003 | $-2.2·10^{-7}$ to $2.8·10^{-7}$ | 0.37-0.82 | 0.99 (0.99-1.00) | 9.8 (8-14) |
| 8 | $CH_4 = aR^b + cH(aR^b) + K$ | V4a | 719 | 0.001-0.003 | $-5.6·10^{-7}$ to $1.3·10^{-7}$ | 0.37-0.84 | 1.00 (0.99-1.00) | 10.4 (8-15) |
| 9a | As No. 8 with equation V4b to determine V0. | V4b | 719 | 0.001-0.003 | $-7.8·10^{-7}$ to $1.4·10^{-6}$ | 0.37-0.84 | 1.00 (0.99-1.00) | 10.4 (8-15) |
| 9b | As No. 9a with lower max CH4 level. | V4b | 50 | 0.007-0.014 | $-4.1·10^{-8}$ to $8.1·10^{-8}$ | 0.16-0.33 | 0.98 (0.96-0.99) | 2.1 (2-3) |





| | | | | | | | | |
|---|---|---|---|---|---|---|---|---|
| 10a | As No 8. Parameters *a*, *b*, *c* and *K* estimated from relationships with parameters in V4b; see text. | V4b | 719 | 0.001-0.012 | -108 to 1.1 | 0.39-1.62 | 1.00 (0.99-1.00) | 74 (18-150) |
| 10b | As No 8. Parameters *a*, *b*, *c* and *K* estimated from relationships with parameters in V5b; see text. | V5b | 719 | 0.001-0.024 | -122 to 1.9 | 0.43-2.80 | 0.99 (0.96-1.00) | 88 (20-154) |
| 10c | As No. 10a with lower max $CH_4$ level. | V4b | 50 | 0.006-0.021 | -51 to -14 | 0.30-0.87 | 0.98 (0.96-0.98 | 28 (21-35) |
| 11a | As No. 8. Parameters *a*, *b*, *c*, and *K, derived* from 6-7 minimum $V_L$ values at different *H*, and 20 samples at random $CH_4$ levels between 3 and 500 ppm.** | V4b | 719 | 0.002-0.004 | -6.5 to 7.1 | 0.41-0.96 | 1.00 (0.99-1.00) | 13 (8.8-20) |
| 11b | As No. 11a with the 20 random samples at $CH_4$ levels between 3 and 50 ppm.** | V4b | 50 | 0.008-0.017 | -0.7 to 0.5 | 0.17-0.41 | 0.97 (0.95-0.98) | 2.5 (2-3) |

*Minimum and maximum mean intercepts for the group of 20 sensors. The confidence interval around the mean intercept was ± 1.1 ppm in Model 7 (having lowest RMSE). ** Monte Carlo simulations with 1000 runs generating random data subsets used for deriving the model parameter ranges.






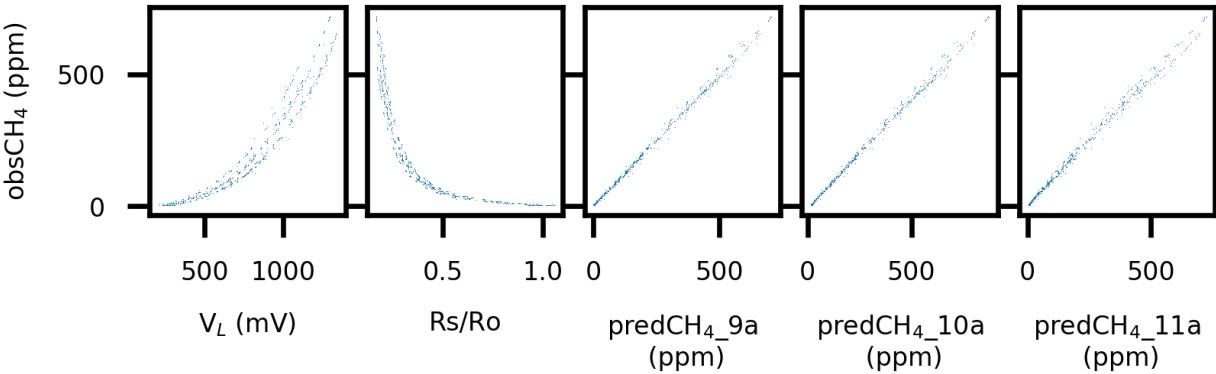

**Figure 1.** Sensor output voltage ($V_L$; mV), $Rs/R_0$ ratio, and predicted CH$_4$ mixing ratio (predCH$_4$; ppm) using Model 9a, 10a and 11a in Table 2, respectively, versus observed CH$_4$ mixing ratio (obsCH$_4$; ppm), for one of the studied sensors. See text for details and Figure S1 for similar graphs regarding all sensors.






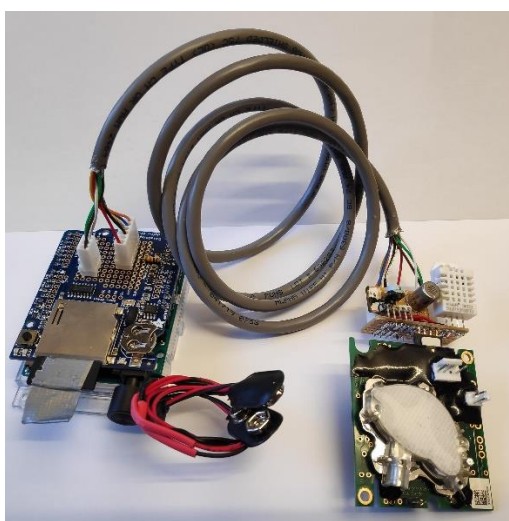

**Figure 2.** Photo of the CH4/CO2/RH/T logger described in the supplementary information.