# Peer review of "Technical note: Facilitating the use of low-cost methane (CH4) sensors in flux chambers – calibration, data processing, and an open source make-it-yourself logger"

_Biogeosciences, 2019_

## Referee Comment (RC1) · Anonymous Referee #1 · 12 Feb 2020

In this paper Bastviken and colleagues present details on a low cost sensor for measuring CH4 fluxes in chambers. They also describe a low-cost open source logger. There is a growing interest in development and use of low cost sensors for measuring key biogeochemical processes, and this paper describes a useful new sensor to add to the growing list. The paper is focused on calibration and data processing using this new sensor setup.

One thing that I would have liked to have seen was some real word data using the system - that is a demonstration of field based flux measurements. It is under field

conditions where the utility of the sensor needs to be proved. In saying this, I think the paper as it stands is publishable and will make a useful contribution to the scientific literature. The detailed calibration experiments will be extremely useful for the community working on developing similar systems. look forward to seeing "real-world" data collected by the sensor system in the future.

---

## Referee Comment (RC2) · Anonymous Referee #2 · 20 Feb 2020

The technical note describes comprehensive laboratory tests of a low-cost methane sensor for potential application in flux chamber measurements in aquatic systems. The results are of high relevance for enhancing the spatial and temporal resolution of methane flux measurements and for improving our understanding of their environmental controls. The results and conclusion are comprehensible and well backed-up by data. Moreover, the note provides detailed instructions and procedures for implementing and calibrating such sensor in future applications. It is generally well written and presented. I have a few detailed, minor suggestions for improving the clarity of the

note, which are listed below:

- the authors generally refer to "methane levels", which are reported in ppm. I suggest to clarify at some point what exactly is meant by this – mol fraction, mixing ratio?

- similarly, the authors state that the tested sensor is measuring "methane concentration", however, all results are reported in ppm. Could it be that the sensor responds to the abundance of CH4 molecules, rather than the mixing ratio (as it has been found for humidity in the present manuscript)? The difference between a concentration (as mols or mass per volume) and mol fraction is the temperature and pressure dependence (as it could be described by the ideal gas law).

- the range of relative humidity used in the experiments was 18-70%. Depending on deployment time, humidity in a flux chamber can become much higher than this. Could you add a remark on sensor performance at higher humidity?

- the proposed Arduino-based logger includes a humidity/temperature sensor in addition to the CO2 sensor, which already includes sensors for both parameters. What is the reason for adding this additional sensor? Does it have higher accuracy?

- line 195: "However, results indicate that the relative change of CH4 levels over time, which is the core of flux chamber measurements, can be assessed efficiently with the sensors if calibrated properly." I do not see how this conclusion can be made is this point. Consider adding a justification.

- An important conclusion of the study was that "Sensor-specific calibration is required". Could you add some information about the stability of the calibration obtained for a specific sensor over extended time periods? What calibration intervals would you suggest?

- Table 1: the symbol n is used as a coefficient in equation V1 and also for number of samples. I suggest to use separate symbols. The parameter h (listed in the caption) seems not to be used in the table.

---

## Referee Comment (RC3) · Anonymous Referee #3 · 26 Feb 2020

The authors are to be commended for working toward affordable instrumentation for trace gas measurement. In addition, information provided for construction of a low-cost datalogger can be useful for other environmental measurements as well.

The authors take a primarily empirical approach to the problem of extending the range of usability of a commercial sensor. The goal is to be able to measure low levels that would be of interest in natural ecosystems. Toward this goal they describe a variety of curve-fitting calibrations. It seems possible that many readers will find the most value in knowing how best to apply these sensors, and what ultimate performance can be

achieved. This reviewer suggests that revisions focus mostly on optimum calibration procedures and the perfomance metrics that can be so obtained. The following are some specific comments addressing individual statements in the manuscript.

Line 59 typo (is are)

L 63 It would be helpful to know what mixing ratios were successfully measured

L 77 why were 10 RH sensors used? Was this to provide some averaging?

L 84 How does measured RH compare with the vapor pressure of water at the given temps?

L 89 typo (form) , superfluous 'before'

L 92 some further discussion of time response is called for, especially if some data points are to be removed from the analysis on the basis of delayed responses

L 92 typo (therefor)

L 108 it is unclear why knowing RL is considered challenging, as 1% resistors are routinely available at low cost.

L 123 this reviewer is not in a good position to assess the calibration approaches in detail, as they are mainly empirical and specific to these particular sensors (which this reviewer has not personally used)

L 168 a few more details of the datalogger would be of interest to readers here. What is input voltage range and resolution (e.g. number of significant bits?). What other parameters would someone wanting to use this device in the field want to know? (see also below)

L 176 typo (influence)

L 180 self heating is a very interesting issue. How much power is dissipated at the sensor surface, and would it be expected to produce heating that is significant relative

to the uncertainty with which sensor temperature is constrained/influenced by environmental conditions?

L 220 It would be most helpful if the conclusions stated here were expressed more quantitatively, expressed perhaps in terms of accuracy, reproducibility, and long-term stability.

L 234 Good that code is provided for the datalogger!

Supplement

The datalogger may be of interest to many who plan on building their own field instrumentation. A more detailed circuit diagram, perhaps accompanied by a clear and more explicit image of the physical setup, would be helpful to those not experienced with Arduino.

Fig S1 shows responses over ranges of several hundred ppm. It is suggested to also present data on an expanded scale at the lower end of the usable range.

Fig. S2 what are the units associated with RMSE in this figure. It gives the impression that acceptable results can not be obtained without using 8 or 9 reference samples.

General: what are the authors' observations with regard to aging and long-term stability of these sensors?

---

## Author Comment (AC3) · 4 May 2020

Response to comments by Referee #3

Referee comments are provided in grey and author responses in *blue italic style text*.

The authors are to be commended for working toward affordable instrumentation for trace gas measurement. In addition, information provided for construction of a lowcost datalogger can be useful for other environmental measurements as well. The authors take a primarily empirical approach to the problem of extending the range of usability of a commercial sensor. The goal is to be able to measure low levels that would be of interest in natural ecosystems. Toward this goal they describe a variety of curve-fitting calibrations.

It seems possible that many readers will find the most value in knowing how best to apply these sensors, and what ultimate performance can be achieved. This reviewer suggests that revisions focus mostly on optimum calibration procedures and the perfomance metrics that can be so obtained. The following are some specific comments addressing individual statements in the manuscript.

*We thank Referee #3 for all work and are glad that the efforts towards affordable trace gas instrumentation are seen as valuable. We will try to highlight and clarify our optimization approach further, including that we focused on flux chamber use of these sensors and that this use have different calibration requirements than e.g. use in open air to follow atmospheric mol fractions. In the flux chamber applications, accurate determination of relative changes in gas mol fractions is more important than accuracy in determining absolute mol fraction values. Because of the flux chamber focus, we also calibrated the sensors when positioned in a flux chamber, i.e. at field-like conditions, and we tried to evaluate also simplified calibration procedures as a way to present different acceptable solutions being optimal for field-conditions and various access to laboratory analyses. We are well aware that this is not optimal from the absolute "maximum accuracy and precision" perspective, where it would be desirable to keep a stricter environmental control during calibration. Hence, we need to clarify that we, as also noted by Referee 3, have a more empirical and field-use oriented approach with the aim to facilitate reliable flux chamber measurements by as many sensor users as possible (which is different from assessing the maximum performance of the sensor under stable lab conditions).*

Line 59 typo (is are)

*Will be fixed.*

L 63 It would be helpful to know what mixing ratios were successfully measured

*This range is provided in Table S2 that will be cited here in the revised manuscript.*

L 77 why were 10 RH sensors used? Was this to provide some averaging?

*We had 10 sensors measuring CO2, T and RH in the same chamber as the CH4 sensors to also evaluate our sensor network solution for the CO2/T/RH sensors (separate study). Because we could not link values from any specific CO2/T/RH sensor to each specific CH4 sensors (they all shared the same chamber volume) we decided to just average their values and use this average in the evaluation of the CH4 sensors. We will clarify this in the revised manuscript.*

L 84 How does measured RH compare with the vapor pressure of water at the given temps?

*We used absolute humidity in the unit of ppm water vapor, corresponding to vapor pressure in µatm. The study covered a humidity range from 9000 to 35 000 ppm water vapor. At temperatures below 20*

*°C the RH was usually 50 - 70 %, while at temperatures > 20 °C, RH ranged from 18 to 60 %. Our result indicates that the absolute humidity was the most important factor, followed by temperature, and therefore we think the absolute ranges in these variables matters more than the RH values for the sensor tests. We will try to clarify this in the revised manuscripts.*

L 89 typo (form) , superfluous 'before'
*Will be fixed.*

L 92 some further discussion of time response is called for, especially if some data points are to be removed from the analysis on the basis of delayed responses

*Gas concentrations in the chamber with the sensors were impacting the sensors immediately and recorded at their respective logging frequency. It took some time for this gas to pass of the tubing and flush the measurements cell in the reference instrument. Hence, the reference instrument responded with some delay. When the concentrations were relatively stable or changed at slow constant rates this could be corrected for by considering the time offset. However, when concentrations in the sensor chamber changed rapidly relative to the time off-set, e.g. right after $CH_4$ was added to the chamber or when the chamber was ventilated to reduce gas concentrations, it was not possible to correct for the time off-set. The reference instrument measurement cell was simply large enough to be influenced by $CH_4$ from the chamber over a certain time period (the measurement cell residence time), and if the concentration change in the chamber is more rapid, the data from the reference instrument and sensors become incomparable and need to be omitted to not bias the calibration. We tried to explain this in the previous sentence but will consider further clarifications when revising the manuscript.*

L 92 typo (therefor)

*Will be fixed.*

L 108 it is unclear why knowing RL is considered challenging, as 1% resistors are routinely available at low cost.

*Referee #3 is correct that most of the variability between sensors are likely to regard the active sensor surface characteristics and that variability regarding RL is less likely. We just did not want to exclude this possibility, but given this comment, we will consider removing RL from this sentence.*

L 123 this reviewer is not in a good position to assess the calibration approaches in detail, as they are mainly empirical and specific to these particular sensors (which this reviewer has not personally used)

*Noted. Many thanks for good and important comments overall.*

L 168 a few more details of the datalogger would be of interest to readers here. What is input voltage range and resolution (e.g. number of significant bits?). What other parameters would someone wanting to use this device in the field want to know? (see also below)

*The Arduino based datalogger has an input voltage range of 7-12 V and a resolution of 10 bits. There is good documentation regarding logger board specifications on internet (e.g. https://store.arduino.cc/arduino-uno-rev3 and https://learn.adafruit.com/adafruit-data-logger-shield). We will check and make sure information about such links are available in the supplement.*

L 176 typo (influence)

*Will be fixed.*

L 180 self heating is a very interesting issue. How much power is dissipated at the sensor surface, and would it be expected to produce heating that is significant relative to the uncertainty with which sensor temperature is constrained/influenced by environmental conditions?

*The heating power consumption of the CH4 sensor is approximately 280 mW. This was not enough to notably change the temperature relative to other factors in the flux chambers tested so far. Not even under the experimental conditions when we had 20 sensors in a chamber with 7 L gas volume did we see any considerable heating effects.*

L 220 It would be most helpful if the conclusions stated here were expressed more quantitatively, expressed perhaps in terms of accuracy, reproducibility, and long-term stability.

*We considered this carefully and in principle agree. However, given that several different calibration models gave acceptable results and may be optimal for different conditions and different users, we would have to repeat rather large amounts of information to provide this information properly. In turn, this would lead to a rather long conclusions section. To try to resolve this situation we refer readers to Table 2 for quantitative information regarding different calibration strategies.*

*For long-term stability, we will add such information to the manuscript. Please see the response to the last comment below.*

L 234 Good that code is provided for the datalogger!

*Thanks.*

Supplement
The datalogger may be of interest to many who plan on building their own field instrumentation. A more detailed circuit diagram, perhaps accompanied by a clear and more explicit image of the physical setup, would be helpful to those not experienced with Arduino.

*We tried to provide a clear wiring diagram and illustrative images. We will consider how we can improve the visual descriptions of the system further in the revised supplement.*

Fig S1 shows responses over ranges of several hundred ppm. It is suggested to also present data on an expanded scale at the lower end of the usable range.

*We will add a similar figure covering the lower end of the range in the revised supplement.*

Fig. S2 what are the units associated with RMSE in this figure. It gives the impression that acceptable results can not be obtained without using 8 or 9 reference samples.

*The RMSE units are ppm, which will be clarified in the revised manuscript. The impression of Referee #3 is correct. A minimum of 8-9 reference samples are needed for acceptable results. In the text we choose to highlight that 20 reference samples would be preferable to reach even lower RMSE levels, but we will consider to relax this conclusion to 10 reference samples given Fig. S2 as suggested by Reviewer #3.*

General: what are the authors' observations with regard to aging and long-term stability of these sensors?

*We could not observe any tendencies of ageing during our studies so far, but we have added information about this based on other studies. van den Bossche et al., (2017; cited in manuscript) found no drift over 31 days. Eugster et al (2019) studied results from a very similar type of sensor used for outdoor measurements over 7 years and concluded that the drift was in the order of 4–6 ppb/yr and the variability drifted by –0.24%/yr. We will add this information in the revised study to address the drift question.*

*Reference:*
*Eugster, W., Laundre, J., Eugster, J., and Kling, G. W.: Long-term reliability of the Figaro TGS 2600 solid-state methane sensor under low Arctic conditions at Toolik lake, Alaska, Atmos. Meas. Tech. Discuss., https://doi.org/10.5194/amt-2019-402, in review, 2019.*

---

## Author Comment (AC4) · 4 May 2020

The comment was uploaded in the form of a supplement:
https://www.biogeosciences-discuss.net/bg-2019-499/bg-2019-499-AC4-supplement.pdf

---

## Author Response (AR1)

Dear Dr Ji-Hyung Park,

Many thanks for all work with our manuscript bg-2019-499, entitled *Technical note: Facilitating the* use of low-cost methane (CH4) sensors in flux chambers – calibration, data processing, and an open source make-it-yourself logger

We are very grateful to all involved, including the three reviewers, for the very supportive comments that substantially improved the manuscript. Please find below point-by-point comments to all review comments.

With regards to the specific Editorial report comments, we have, as requested by Reviewer #1, added real data from flux chamber measurements on a lake as a figure in the Supplement. Regarding optimum calibration procedures, addressed by Reviewer #3, we added text to clarify that advanced optimum calibration under strictly controlled standardized conditions, which of course is better for sensor comparisons and some applications, were not the aim in this study because we wanted to provide good-enough calibration solutions for the widest possible range of users, most of which does not have access to laboratories where well defined standard calibration conditions can be created. This is key for cost efficiency as the sensors require individual and repeated calibrations over their life time.

As requested we uploaded our manuscript and Supplement files file with changes marked (form MS Word Track Changes converted to pdf). We also have a files without track changes ready to send if desired as we could not find space for submitting two manuscript files in the web form. All the line numbers provided in our response below refer to the submitted files with track changes.

We are happy to respond to any emerging questions and again thank you and others involved for all the work with our manuscript.

Best regards, David Bastviken

**Response to comments by Referee #1**

**Referee comments are provided in grey and author responses in *blue italic style text*.**

In this paper Bastviken and colleagues present details on a low cost sensor for measuring CH4 fluxes in chambers. They also describe a low-cost open source logger. There is a growing interest in development and use of low cost sensors for measuring key biogeochemical processes, and this paper describes a useful new sensor to add to the growing list. The paper is focused on calibration and data processing using this new sensor setup.

One thing that I would have liked to have seen was some real word data using the system - that is a demonstration of field based flux measurements. It is under field conditions where the utility of the sensor needs to be proved. In saying this, I think the paper as it stands is publishable and will make a useful contribution to the scientific literature. The detailed calibration experiments will be extremely useful for the community working on developing similar systems. look forward to seeing "real-world" data collected by the sensor system in the future.

**We thank Referee #1 for all work and are glad that our work is considered valuable and important. We have provided examples of real field based measurements in the revised Supplement (Fig S4).**

**Response to comments by Referee #2**

**Referee comments are provided in grey and author responses in *blue italic style text*.**

The technical note describes comprehensive laboratory tests of a low-cost methane sensor for potential application in flux chamber measurements in aquatic systems. The results are of high relevance for enhancing the spatial and temporal resolution of methane flux measurements and for improving our understanding of their environmental controls. The results and conclusion are comprehensible and well backed-up by data. Moreover, the note provides detailed instructions and procedures for implementing and calibrating such sensor in future applications. It is generally well written and presented. I have a few detailed, minor suggestions for improving the clarity of the note, which are listed below:

**We thank Referee #2 for all work and are glad that the manuscript is found relevant and of value in several ways. The detailed comments are addressed below.**

- the authors generally refer to "methane levels", which are reported in ppm. I suggest to clarify at some point what exactly is meant by this – mol fraction, mixing ratio?

**We have tried to clarify this in the revised manuscript. In many cases, mol fraction replaced levels when the text was about CH4 in ppm units (many lines throughout the whole text). Levels were kept for text about multiple variables or when discussing relative changes rather than absolute mol fraction numbers.**

- similarly, the authors state that the tested sensor is measuring "methane concentration", however, all results are reported in ppm. Could it be that the sensor responds to the abundance of CH4 molecules, rather than the mixing ratio (as it has been found for humidity in the present manuscript)? The difference between a concentration (as mols or mass per volume) and mol fraction is the temperature and pressure dependence (as it could be described by the ideal gas law).

Referee #2 is correct that the sensors respond to the abundance of the molecules interacting with the sensor surface which is proportional to the mol fraction. The conversion from mol fraction to concentration (mol or mass per volume) via the ideal gas law correct for some of the temperature and pressure effects. On top of this, there can be extra temperature effects on sensor responses if the temperature influence the electronic component performance or the sensor surface characteristics. In this case water vapor also interact with the sensor surface, which explains the need for extra measures to correct for humidity. We have tried to clarify these aspects on row 101 and 227-229.

- the range of relative humidity used in the experiments was 18-70%. Depending on deployment time, humidity in a flux chamber can become much higher than this. Could you add a remark on sensor performance at higher humidity?

The highest absolute humidity reached in our experiments were in the order of 35 000 ppm H20. This corresponds to approximately 100% humidity at 26 °C which is valid for many conditions. Unfortunately we do not dare to make any statements regarding sensor performance outside our test range, so there will be a need for tests at higher humidity in the future, in e.g. tropical conditions. The highest absolute humidity covered have been clarified in the revised manuscript (lines 86-88).

- the proposed Arduino-based logger includes a humidity/temperature sensor in addition to the CO2 sensor, which already includes sensors for both parameters. What is the reason for adding this additional sensor? Does it have higher accuracy?

We added this extra RH/T sensor just to have a backup to the RH and T sensors at the CO2 sensor. This was done because the CH4 sensor data evaluation (in contrast to the CO2 sensor data) is completely dependent on reliable RH and T information to calculate absolute humidity. We now explain this in the revised supplementary material (section on Hardware setup).

- line 195: "However, results indicate that the relative change of CH4 levels over time, which is the core of flux chamber measurements, can be assessed efficiently with the sensors if calibrated properly." I do not see how this conclusion can be made is this point. Consider adding a justification.

We agree that this sentence needs clarification. The justification is based on data in Table 2 where the SE and R2 for the slope versus the SE for the intercept of the calibration equations indicated that the relative change in sensor response to increasing CH4 mol fraction was rather consistent and accurate (more accurate than the absolute mol fraction derived). We have clarified this (lines 201-205).

- An important conclusion of the study was that "Sensor-specific calibration is required". Could you add some information about the stability of the calibration obtained for a specific sensor over extended time periods? What calibration intervals would you suggest?

This is a very important question. We do not yet have data from long-term studies to present, but other studies have addressed this. van den Bossche et al., (2017; cited in manuscript) found no drift over 31 days. Eugster et al (2019) studied results from a very similar type of sensor used for outdoor measurements over 7 years and concluded that the drift was in the order of 4–6 ppb/yr and the variability drifted by –0.24%/yr. We have added this information in the revised study (lines 239-244).

**Reference:**

*Eugster, W., Laundre, J., Eugster, J., and Kling, G. W.: Long-term reliability of the Figaro TGS 2600 solid-state methane sensor under low Arctic conditions at Toolik lake, Alaska, Atmos. Meas. Tech. Discuss., https://doi.org/10.5194/amt-2019-402, in review, 2019.*

- Table 1: the symbol n is used as a coefficient in equation V1 and also for number of samples. I suggest to use separate symbols. The parameter h (listed in the caption) seems not to be used in the table.

Thanks for such detailed reading! We have changed to "N" for number of samples. "h" is used once in Model V2 in Table 1.

**Response to comments by Referee #3**

**Referee comments are provided in grey and author responses in *blue italic style text*.**

The authors are to be commended for working toward affordable instrumentation for trace gas measurement. In addition, information provided for construction of a lowcost datalogger can be useful for other environmental measurements as well. The authors take a primarily empirical approach to the problem of extending the range of usability of a commercial sensor. The goal is to be able to measure low levels that would be of interest in natural ecosystems. Toward this goal they describe a variety of curve-fitting calibrations.

It seems possible that many readers will find the most value in knowing how best to apply these sensors, and what ultimate performance can be achieved. This reviewer suggests that revisions focus mostly on optimum calibration procedures and the performance metrics that can be so obtained. The following are some specific comments addressing individual statements in the manuscript.

We thank Referee #3 for all work and are glad that the efforts towards affordable trace gas instrumentation are seen as valuable. We have tried to highlight and clarify our approach further, including that we focused on flux chamber use of these sensors and that this use have different calibration requirements than e.g. use in open air to follow atmospheric mol fractions (lines 225-237). In the flux chamber applications, accurate determination of relative changes in gas mol fractions is more important than accuracy in determining absolute mol fraction values. Because of the flux chamber focus, we also calibrated the sensors when positioned in a flux chamber, i.e. at field-like conditions, and we tried to evaluate also simplified calibration procedures as a way to present different acceptable solutions being optimal for field-conditions and various access to laboratory analyses. We are well aware that this is not optimal from the absolute "maximum accuracy and precision" perspective, where it would be desirable to keep a stricter environmental control during calibration. Hence, we need to clarify that we, as also noted by Referee 3, have a more empirical and field-use oriented approach with the aim to facilitate reliable flux chamber measurements by as many sensor users as possible (which is different from assessing the maximum performance of the sensor under stable lab conditions).

Line 59 typo (is are)

**Fixed.**

L 63 It would be helpful to know what mixing ratios were successfully measured

**The mixing rations of the mentioned reference have been added to the text.**

L 77 why were 10 RH sensors used? Was this to provide some averaging?

We had 10 sensors measuring CO2, T and RH in the same chamber as the CH4 sensors to also evaluate our sensor network solution for the CO2/T/RH sensors (separate study). Because we could not link values from any specific CO2/T/RH sensor to each specific CH4 sensors (they all shared the same chamber volume) we decided to just average their values and use this average in the evaluation of the CH4 sensors. We have clarified this in the revised manuscript (lines 78-79 and 88-90).

L 84 How does measured RH compare with the vapor pressure of water at the given temps?

We used absolute humidity in the unit of ppm water vapor, corresponding to vapor pressure in  $\mu$ atm. The study covered a humidity range from 9000 to 35 000 ppm water vapor. At temperatures below 20  $^{\circ}$  the RH was usually 50 - 70 %, while at temperatures > 20  $^{\circ}$ , RH ranged from 18 to 60 % (clarified on lines 86-88). Our result indicates that the absolute humidity was the most important factor, followed by temperature, and therefore we think the absolute ranges in these variables matters more than the RH values for the sensor tests.

L 89 typo (form) , superfluous 'before' *Fixed.*

L 92 some further discussion of time response is called for, especially if some data points are to be removed from the analysis on the basis of delayed responses

Gas concentrations in the chamber with the sensors were impacting the sensors immediately and recorded at their respective logging frequency. It took some time for this gas to pass of the tubing and flush the measurements cell in the reference instrument. Hence, the reference instrument responded with some delay. When the concentrations were relatively stable or changed at slow constant rates this could be corrected for by considering the time offset. However, when concentrations in the sensor chamber changed rapidly relative to the time off-set, e.g. right after CH4 was added to the chamber or when the chamber was ventilated to reduce gas concentrations, it was not possible to correct for the time off-set. The reference instrument measurement cell was simply large enough to be influenced by CH4 from the chamber over a certain time period (the measurement cell residence time), and if the concentration change in the chamber is more rapid, the data from the reference instrument and sensors become incomparable and need to be omitted to not bias the calibration. We have now tried to clarify this further (line 93-96).

**L 92 typo (therefor)**

**Fixed.**

L 108 it is unclear why knowing RL is considered challenging, as 1% resistors are routinely available at low cost.

**Referee #3 is correct that most of the variability between sensors are likely to regard the active sensor surface characteristics and that variability regarding RL is less likely. We just did not want to exclude this possibility, but given this comment, we have removed RL from this sentence.**

L 123 this reviewer is not in a good position to assess the calibration approaches in detail, as they are mainly empirical and specific to these particular sensors (which this reviewer has not personally used)

**Noted. Many thanks for good and important comments overall.**

L 168 a few more details of the datalogger would be of interest to readers here. What is input voltage range and resolution (e.g. number of significant bits?). What other parameters would someone wanting to use this device in the field want to know? (see also below)

The Arduino based datalogger has an input voltage range of 7-12 V and a resolution of 10 bits. This has been added to the text (line 177).

There is good documentation regarding logger board specifications on internet (e.g. https://store.arduino.cc/arduino-uno-rev3 and https://learn.adafruit.com/adafruit-data-logger-shield, which has been added to the supplement.

**L 176 typo (influence)**

**Checked but could not find the error.**

L 180 self heating is a very interesting issue. How much power is dissipated at the sensor surface, and would it be expected to produce heating that is significant relative to the uncertainty with which sensor temperature is constrained/influenced by environmental conditions?

**The heating power consumption of the CH4 sensor is approximately 280 mW. This was not enough to notably change the temperature relative to other factors in the flux chambers tested so far. Not even**

under the experimental conditions when we had 20 sensors in a chamber with 7 L gas volume did we see any considerable heating effects.

**We added brief information about this (lines 186-187)**

L 220 It would be most helpful if the conclusions stated here were expressed more quantitatively, expressed perhaps in terms of accuracy, reproducibility, and long-term stability.

We considered this carefully and in principle agree. However, given that several different calibration models gave acceptable results and may be optimal for different conditions and different users, we would have to repeat rather large amounts of information to provide this information properly. In turn, this would lead to a rather long conclusions section. To try to resolve this situation we refer readers to Table 2 for quantitative information regarding different calibration strategies.

For long-term stability, we have added such information to the manuscript (lines 239-244). Please see the response to the last comment below.

L 234 Good that code is provided for the datalogger!

**Thanks.**

**Supplement**

The datalogger may be of interest to many who plan on building their own field instrumentation. A more detailed circuit diagram, perhaps accompanied by a clear and more explicit image of the physical setup, would be helpful to those not experienced with Arduino.

We tried to provide a clear wiring diagram and illustrative images. We realize that the varnish on the CO2 sensor used for moisture protection makes it hard to see all details, but we in the revision tried to improve the visual descriptions of the system by enlarging all images and also provide a link to the CO2 sensor producer web page where very detailed information about the sensor board and connection points can be found.

Fig S1 shows responses over ranges of several hundred ppm. It is suggested to also present data on an expanded scale at the lower end of the usable range.

A similar figure covering the lower end of the range have been added to the revised supplement (new Figure S2).

Fig. S2 what are the units associated with RMSE in this figure. It gives the impression that acceptable results can not be obtained without using 8 or 9 reference samples.

The RMSE units are ppm, which is now clarified in the revised manuscript (Table 2). The impression of Referee #3 is correct. A minimum of 8-9 reference samples are needed for acceptable results. In the text we choose to highlight that 20 reference samples would be preferable to reach even lower RMSE levels, but we have relaxed this conclusion to 10 - 20 reference samples given Fig. S2 as suggested by Reviewer #3.

General: what are the authors' observations with regard to aging and long-term stability of these sensors?

We could not observe any tendencies of ageing during our studies so far, but we have added information about this based on other studies. van den Bossche et al., (2017; cited in manuscript)

found no drift over 31 days. Eugster et al (2019) studied results from a very similar type of sensor used for outdoor measurements over 7 years and concluded that the drift was in the order of 4–6 ppb/yr and the variability drifted by –0.24%/yr. We have add this information in the revised study to address the drift question (lines 239-244).

Reference:

[revised manuscript text omitted]

370 \*Minimum and maximum mean intercepts for the group of 20 sensors. The confidence interval around the mean intercept was ± 1.1 ppm in Model 7 (having lowest RMSE). \*\* Monte Carlo simulations with 1000 runs generating random data subsets used for deriving the model parameter ranges.

**Figure 1.** Sensor output voltage ( $V_L$ ; mV),  $Rs/R_0$  ratio, and predicted CH4 mixing ratio (predCH4; ppm) using Model 9a, 10a and 11a in Table 2, respectively, versus observed CH4 mixing ratio (obsCH4; ppm), for one of the studied sensors. See text for details and Figures S1 and S2 for similar graphs regarding all sensors.